# The Effect of the Position of a Phenyl Group on the Luminescent and TNP-Sensing Properties of Cationic Iridium(III) Complexes

**DOI:** 10.3390/s25030839

**Published:** 2025-01-30

**Authors:** Xiaoran Yang, Jiahao Du, Rui Cai, Chun Liu

**Affiliations:** State Key Laboratory of Fine Chemicals, Frontier Science Center for Smart Materials, School of Chemical Engineering, Dalian University of Technology, Linggong Road 2, Dalian 116024, China; yxiaoran@mail.dlut.edu.cn (X.Y.); dujiahao@mail.dlut.edu.cn (J.D.); cairui@dlut.edu.cn (R.C.)

**Keywords:** cationic Ir (III) complexes, 2,4,6-trinitrophenol, AIPE properties, photo-induced electron transfer

## Abstract

Three cationic Ir(III) complexes, **1**, **2**, and **3**, were successfully synthesized and characterized by tuning the position of a phenyl group at the pyridyl moiety in 2-phenylpyridine. All three complexes exhibited typical aggregation-induced phosphorescence emission (AIPE) properties in CH_3_CN/H_2_O. The AIPE property was further utilized to achieve the highly sensitive detection of 2,4,6-trinitrophenol (TNP) in aqueous media with low limit of detection (LOD) values of 164, 176, and 331 nM, respectively. This suggests that the different positions of the phenyl group influence the effectiveness of **1**, **2**, and **3** in the detection of TNP. In addition, **1**, **2**, and **3** showed superior selectivity and anti-interference properties for the detection of TNP and were observed to have the potential to be used to detect TNP in practical applications. The changes in the luminescence lifetime and UV-Vis absorption spectra of **1**, **2**, and **3** before and after the addition of TNP indicate that the corresponding quenching process is a combination of static and dynamic quenching. Additionally, the proton nuclear magnetic resonance spectra and results of spectral studies show that the detection mechanism is photo-induced electron transfer (PET).

## 1. Introduction

Nitroaromatic compounds are widely used in explosives, as rapid and sensitive detection of nitroaromatic compounds is required for homeland and national security [1,2]. One of these compounds, 2,4,6-trinitrophenol (TNP), is not only employed in nitro explosives but also in many industrial operations such as the production of dyes, pyrotechnics, matches, glass, leather, rocket fuel, and batteries [3,4,5]. However, due to the widespread use of TNP, it may be released into the environment during synthesis, transportation, and disposal, where it exists in the form of ions in the air, water, and soil, potentially causing serious harm to humans. For example, it causes contact dermatitis, conjunctivitis, and bronchitis, among other conditions [6,7,8]. Therefore, the development of highly sensitive and selective methods for the detection of TNP in aqueous media is of great significance for human health and environmental protection.

Nowadays, several analytical techniques, such as high-performance liquid chromatography [9], Raman spectroscopy [10,11], X-ray analysis [12], cyclic voltammetry [13], and electrochemistry [14], have been used for the detection of nitroaromatic explosive compounds, but these techniques are expensive, time-consuming, and bulky. In contrast, photoluminescence has proved to be an ideal analytical tool for trace explosive detection due to its high sensitivity, simplicity, and real-time monitoring capability [15,16,17,18]. However, common luminescent materials tend to produce weak luminescence due to aggregation-caused quenching (ACQ) when in a solid or aggregated state, limiting the detection of TNP in aqueous media. Notably, the concept of aggregation-induced emission (AIE) first proposed by Tang and coworkers effectively addresses the limitation imposed by the ACQ phenomenon on the application of luminescent materials [19]. Consequently, many compounds with AIE properties have been designed and synthesized for TNP detection [20,21,22,23].

To date, the most widely accepted mechanism for the AIE phenomenon is the restriction of intramolecular motion (RIM), in which luminescent substances are usually modified with propeller- or rotor-like substituents to activate AIE properties [24]. Shortly after the AIE concept was discovered, Manimaran and colleagues reported aggregation-induced phosphorescence emission (AIPE) properties in metal complexes [25]. Recently, several metal complexes have been reported to show excellent AIPE properties [26,27]. Compared with fluorescent light sources, phosphorescent light sources have higher luminescence efficiency and good optical and thermal stability. As phosphorescent light sources, Ir(III) complexes are the subject of an increasing amount of attention due to their unique photophysical properties in the fields of photooxidation reduction [28], luminescent probes [29], and photodynamic therapy [30]. Therefore, the development of Ir(III) complexes with AIPE properties in order to detect TNP has become a hot research topic.

2-Phenylpyridine, a unique C^N-type ligand, can be modified to significantly alter the properties of Ir(III) complexes. The phenyl group, as a freely rotatable *π*-planar group, can change the photophysical and application properties of the metal complexes by altering its substitution position on the cyclometalating ligands. Previously, our group used 2-phenylbenzothiazole derivatives as cyclometalating ligands and introduced a phenyl group [31]; as a result, the AIE properties of Ir(III) complexes were regulated and enhanced. We suggested that in a dilute solution, the rotatable phenyl group is active and acts as a relaxation channel for excited-state inactivation. In the aggregated state, the phenyl group is confined due to physical constraints, which block the nonradiative pathway, thus allowing the exciton to decay radiatively. In 2023, Di and coworkers activated the AIE property of Ir(III) complexes by introducing *N*-phenylcarbazole groups on the parent Ir(III) complexes (of note, the synergistic effect of phenyl and carbazole significantly improves the rotational properties of the substituent groups), thus improving the luminescence properties in the aggregated state [32]. In 2016, Zhang and coworkers [33] designed and synthesized four additional Ir(III) complexes by introducing phenyl and aldehyde groups to the ligands of the Ir(III) complexes, respectively. Combining the quantum yield and electroluminescent (EL) performance, the introduction of the phenyl group led to better EL performance and a larger quantum yield compared with the introduction of electron-withdrawing groups, and the results suggest that the introduction of a phenyl group may have improved the performance of the complexes. Therefore, we designed and synthesized Ir(III) complexes **1**–**3** using phenyl-modified 2-phenylpyridine derivatives as cyclometalating ligands and 2,2’-bipyridine in a flexible conformation as the auxiliary ligand (Figure 1, Appendix A). The luminescent properties of **1**–**3** in CH_3_CN/H_2_O were systematically investigated, and the effects of the position of the phenyl group on the sensitivity and selectivity of **1**–**3** for the detection of TNP in aqueous media were probed. In addition, the practical application of **1**–**3** for the detection of TNP in a variety of common water samples was explored.

## 2. Materials and Methods

First, stock solutions of CH_3_CN for **1**, **2**, and **3** (100 μM) were prepared. Then, 300 μL of the stock solution was collected in a quartz cuvette, and 3 mL (10 μM) of samples with different water contents were prepared by adding appropriate volumes of CH_3_CN and deionized water into the stock solution. The fluorescence emission and UV-Vis absorption spectra of these samples were recorded. Samples of **1**, **2**, and **3** (10 μM) were prepared in 200 mL volumetric flasks with 60% water content, and 3 mL of each sample was collected in a quartz cuvette. Additionally, the emission spectra of the 11 blank samples were recorded for calculation of the standard deviation *σ* (Appendix A). TNP solutions were prepared at concentrations ranging from 0.1 to 50 mM in CH_3_CN/H_2_O with 60% water content. The emission spectra were recorded by adding 30 μL of TNP solutions of different concentrations to a cuvette containing 3 mL of the complex sample. In order to perform selectivity and anti-interference experiments for the detection of TNP, we recorded emission spectra after adding 30 μL of different analytes (20 mM of 1,3-dinitrobenzene (1,3-DNB), nitrobenzene (NB), *p*-cresol, 4-methoxyphenol (MEHQ), phenol, and *m*-nitrophenol (3-NP)) and different ionic compounds (20 mM of CaCl_2_, AlCl_3_, SnCl_2_, NiCl_2_, ZnCl_2_, CoCO_3_, KBr, KF, CuSO_4_, and CH_3_COONa) to 3 mL samples of complexes, respectively. Another 30 μL of TNP solution at a concentration of 20 mM was added to the above samples, and the emission spectra were recorded again. In order to investigate the practical application capacity of **1**, **2**, and **3** for the detection of TNP in a variety of common water samples, different common water samples (tap water from the laboratory of Dalian University of Technology, river water from the Lingshui River, and seawater and snow water from Xinghai Bathing Beach, Dalian), instead of deionized water, were prepared for samples of **1**, **2**, and **3**, for which the emission spectra were recorded after the addition of TNP solution (30 μL, 20 mM).

## 3. Results and Discussion

### 3.1. Photophysical and AIPE Properties

The UV–vis absorption and emission spectra of complexes **1**, **2**, and **3** in CH_3_CN are shown in Figure 2. Complexes **1**–**3** have absorption peaks between 220 and 430 nm as well as two major absorption bands. The maximum absorption wavelengths of **2** and **3** following the introduction of phenyl groups at the 4-position and 5-position of the pyridyl moiety of the cyclometalating ligand changed obviously, having red-shifted by 7 nm (**2**) and 24 nm (**3**) compared to **1**, respectively. Similar to other Ir(III) complexes of bipyridine ligands reported in the literature, **1**–**3** exhibit a strong absorption band below 350 nm, which corresponds to the typical ligand-centered (^1^*π*-*π**) transitions. The weak absorption between 380 and 500 nm was attributed to the mixing between metal-to-ligand charge transfer (^1^MLCT, ^3^MLCT) and ligand-centered (^3^*π*-*π**) transitions, which is facilitated by enhanced spin-orbit coupling [34,35,36,37].

As shown in Figure 3, we tested the emission spectra of **1**, **2**, and **3** in CH_3_CN/H_2_O with different water contents. The emission intensity of the complex samples gradually increased as the water content increased from 0% to 60%. The emission intensity of the complex samples was maximized at 60% water content, exhibiting the typical AIPE phenomenon (Figure 3d). This may be attributed to the aggregation of **1** after the water content began to increase, resulting in the locking of the rotatable phenyl portion of the ligand and an increase in emission intensity. This is also evidenced by the dynamic light scattering (DLS) results for **1**–**3** at 60% water content (Appendix A).

### 3.2. Detection of TNP

The AIPE properties exhibited by **1**, **2**, and **3** in CH_3_CN/H_2_O with 60% water content prompted us to use them as phosphorescent materials for the detection of TNP in aqueous media. We further carried out luminescence-quenching experiments with different concentrations of TNP for **1**, **2**, and **3**, respectively. The results show that the emission intensity of the complex samples decreased with the increase in the concentration of added TNP (Figure 4). When the concentration of added TNP was 200 μM, the quenching efficiency of the complex samples reached about 95% (Appendix A).

The quenching constant (*K*_SV_) represents the sensitivity of a probe. We further analyzed the sensitivity of complexes to TNP using the Stern–Volmer equation: *I*_0_/*I* = *K*_SV_[Q] + 1 (where *I*_0_ and *I* represent the emission intensities of the complexes without TNP and after the addition of different concentrations of TNP, respectively, and [Q] represents the molar concentration of TNP). We fitted the concentration of added TNP to *I*_0_/*I*, showing good linear and nonlinear relationships (Figure 5a–c). The Stern–Volmer plots showed good linearity when the TNP concentration was in the range of 0–9 μM. The Stern–Volmer plots were nonlinear at TNP concentrations of 0–300 μM. In the added TNP concentration range of 0–9 μM, we calculated *K*_SV_ values of 2.88 × 10^4^, 2.26 × 10^4^, and 1.87 × 10^4^ M^−1^ for **1**, **2**, and **3**, respectively, via linear fitting.

The limit of detection (LOD) is also an important parameter for judging the nature of a probe. In order to calculate the LOD values of **1**–**3**, a linear plot of the emission intensity of the complexes and the TNP concentration was created. The slope *K* of the linear equation was obtained via linear fitting (Figure 5d–f). The linear equations for the emission intensities of **1**–**3** with respect to the concentration of TNP yielded *K* of 5.30, 6.97, and 5.89 μM^−1^, respectively. Based on the standard deviation *σ* calculated from the above experiments (Appendix A) and the limit-of-detection formula, i.e., LOD = 3*σ*/*K*, the LOD values for **1**–**3** were calculated to be 164, 176, and 331 nM, respectively. These results suggest that **1**, **2**, and **3** can be used as probes for the efficient detection of TNP.

In addition to *K*_SV_ and LOD, selectivity and anti-interference ability are also important indicators for judging the performance of probes. In order to further investigate whether **1**, **2**, and **3** exhibit good selectivity and anti-interference during TNP detection, we selected several common nitro compounds for use in investigating whether **1**, **2**, and **3** have good selectivity, including NB, MEHQ, *p*-cresol, 3-NP, 1,3-DNB, and phenol. The emission intensity of the samples did not change significantly after adding different nitro compounds (20.0 equiv.), indicating that the different nitro compounds hardly produce a quenching effect on **1**, **2**, and **3** (Figure 6a–c). On the basis of the addition of various nitro compound solutions, we continued to add TNP solution (20.0 equiv.), and the quenching efficiencies of the samples of **1**, **2**, and **3** significantly improved, all being around 95% (Figure 6d). Therefore, **1**, **2**, and **3** can realize the selective detection of TNP.

We added various ionic compounds to the complex samples to investigate whether **1**, **2**, and **3** have good anti-interference properties during TNP detection, including CaCl_2_, AlCl_3_, SnCl_2_, NiCl_2_, ZnCl_2_, CoCO_3_, KBr, KF, CuSO_4_, and CH_3_COONa. After adding the different ionic compounds (20.0 equiv.) to samples of **1**, **2**, and **3**, the emission intensity of the samples did not change obviously, indicating that the different ionic compounds had almost no quenching effect on **1**, **2**, and **3** (Figure 7a–c). Based on the results of the addition of various ionic compounds, we continued to add TNP (20.0 equiv.) and found that the various ionic compounds had almost no effect on the luminescence-quenching efficiencies of **1**, **2**, and **3** (Figure 7d). Thus, **1**, **2**, and **3** show excellent anti-interference properties when used to detect TNP.

In order to further evaluate the ability of **1**, **2**, and **3** to detect TNP in common water samples and better promote the practical application of **1**, **2**, and **3**, we selected tap water, river water, seawater, and snow water to replace deionized water. The results show that the shape and intensity of the emission spectra of **1**, **2**, and **3** remained more constant in the common water samples compared to the case for deionized water (Figure 8). The quenching efficiencies of the complex samples in different common water samples were more than 95% after the addition of TNP (20.0 equiv.). Therefore, **1**, **2**, and **3** all have the potential to detect TNP in actual environments.

### 3.3. Sensing Mechanism

The quenching process is categorized into dynamic and static quenching. During static quenching, the probe interacts with the substance to be measured and forms a non-fluorescent ground-state complex, which does not affect the fluorescence lifetime of the probe. In contrast, dynamic quenching is caused by the collision of the probe with the molecules of the substance to be measured, via energy or charge transfer, and the return of the probe from the excited state to the ground state, resulting in luminescence quenching, which will shorten the emission lifetime of the probe [38,39,40]. As shown by the Stern–Volmer curve in Figure 5, the concentration of added TNP showed a good linear and nonlinear relationship with *I*_0_/*I*. This suggests that the luminescence-quenching processes for **1**, **2**, and **3** might be accompanied by both dynamic and static quenching. In order to better understand the quenching mechanism, we recorded the lifetime decay traces of **1**–**3** after adding different concentrations of TNP (Figure 9a–c). After the addition of TNP, the lifetimes of **1**–**3** decreased with an increasing TNP concentration, indicating that dynamic quenching occurred during the quenching process. *τ*_0_/*τ* shows a good linear relationship with [Q], which suggests that dynamic quenching exists regardless of the concentration of TNP added in the low- or high-concentration range (Figure 9d–f).

In order to better understand the mechanism of the quenching process, we investigated the UV-Vis absorption spectra of **1**–**3** after adding different concentrations of TNP (Figure 10a–c). With the increase in the TNP concentration, the absorption peaks of **1** at 263 nm and 310 nm slightly shifted (a similar phenomenon was observed for **2** and **3**), and the results indicate that static quenching also occurred during the quenching process [41]. To further understand the interaction of **1** with TNP, a Job plot was created by testing how the molar content of **1** varied with emission intensity in the mixed system of **1** and TNP (Figure 10d). The inflection point of the Job plot is 0.5, indicating that the chemical-binding ratio of **1** to TNP is 1:1. In addition, the Benesi–Hildebrand plot of **1** with respect to TNP was obtained by fitting (*I*_0_-*I*)^−1^ to [TNP]^−1^ (Figure 10e). (*I*_0_-*I*)^−1^ has a good linear relationship with [TNP]^−1^, indicating that the chemical-binding ratio of **1** to TNP is 1:1, which supports the results of the Job plot [42,43]. As shown in Appendix A, the inflection point of the Job plots is also 0.5, indicating that the chemical binding ratio of **2** and **3** to TNP is also 1:1.

In CH_3_CN/H_2_O with 60% water content, there was no overlap between the UV-Vis absorption spectrum of TNP and the emission spectrum of **1**–**3**, suggesting that no Förster resonance energy transfer occurred during luminescence quenching (Figure 10f) [44]. In the ^1^H NMR spectrum, there is no significant change in the proton signals of **1**–**3** after the addition of TNP, indicating that TNP did not lead to the decomposition of **1**–**3** (Appendix A). These results suggest that **1**–**3** can achieve the highly sensitive detection of TNP mainly via photo-induced electron transfer (PET) (Appendix A) [45].

## 4. Conclusions

In summary, three cationic Ir(III) complexes, **1, 2**, and **3**, were synthesized by introducing a rotatable phenyl group at different positions of the pyridyl moiety of the cyclometalating ligand (Appendix A). **1**, **2**, and **3** exhibit typical AIPE properties in CH_3_CN/H_2_O. The emission spectra and DLS results for **1**–**3** show that the complex molecules aggregate when water content increases. We have utilized the AIPE properties of **1**, **2**, and **3** to achieve the highly sensitive and selective detection of TNP in aqueous media, and it was found that **1** with a phenyl group at the 3-position of the pyridyl moiety of the cyclometalating ligand has a lower detection limit and higher detection efficiency. The results show that different positions of the phenyl group affected the detection efficiency of **1**, **2**, and **3** for TNP. The lifetimes of **1**–**3** gradually decreased with an increase in the concentration of added TNP, which suggests the existence of dynamic quenching in the quenching process. The absorption peaks of the UV-Vis absorption spectra of **1**–**3** were slightly shifted following the addition of different concentrations of TNP, which indicates that there was also static quenching in the quenching process. The detection mechanism was attributed to PET in the proton nuclear magnetic resonance spectra and spectral studies. These findings not only contribute to the design and construction of novel AIPE-active luminescent complexes at the molecular level but also provide useful information for the development of novel phosphorescent materials for TNP detection.

## Figures and Tables

**Figure 1 sensors-25-00839-f001:**
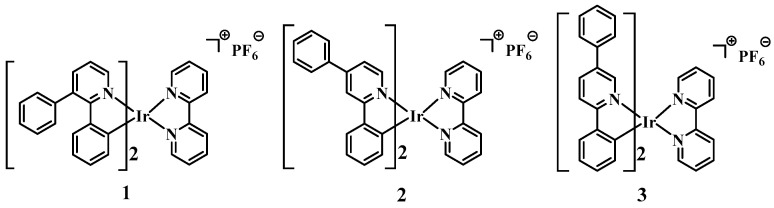
Molecular structures of complexes **1**–**3**.

**Figure 2 sensors-25-00839-f002:**
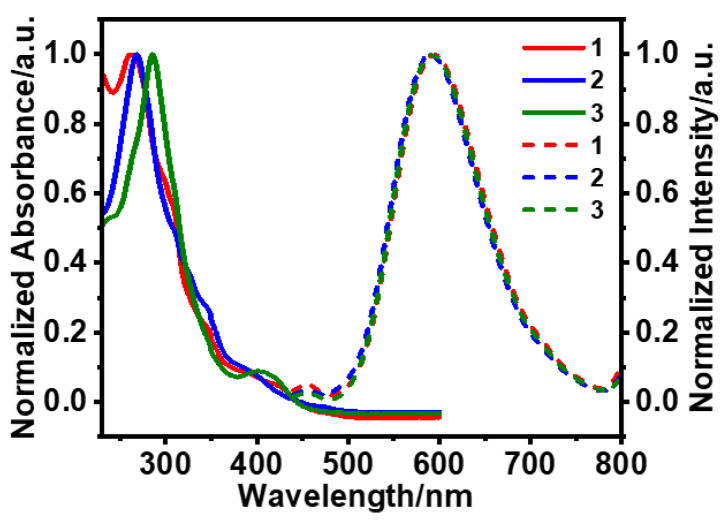
Normalized absorption and emission spectra of **1**, **2**, and **3** in CH_3_CN (solid line: absorption spectrum; dashed line: emission spectrum; excitation wavelength: 400 nm).

**Figure 3 sensors-25-00839-f003:**
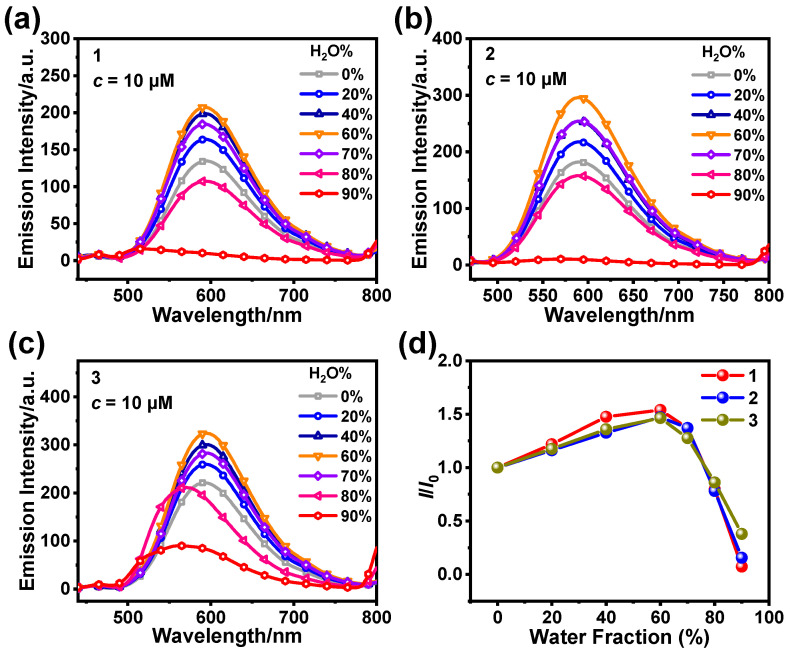
Emission spectra of **1** (**a**), **2** (**b**), and **3** (**c**) in CH_3_CN/H_2_O with various water contents (0–90%) (*c* = 10 µM, *λ*_ex_ = 400 nm). (**d**) Line plots of the ratio of the maximum emission intensity (*I*) of **1**–**3** in CH_3_CN/H_2_O at various water contents to the emission intensity of their monomers (*I*_0_).

**Figure 4 sensors-25-00839-f004:**
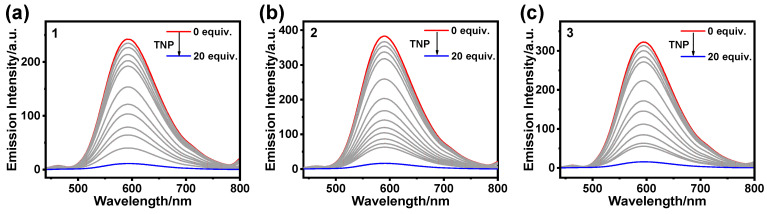
Emission spectra of **1** (**a**), **2** (**b**), and **3** (**c**) (10 μM) in CH_3_CN/H_2_O (*f*_w_ = 60%) as a function of TNP concentration.

**Figure 5 sensors-25-00839-f005:**
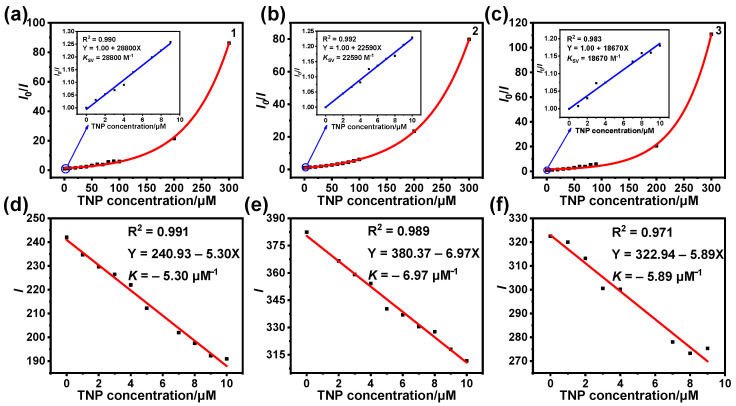
Stern–Volmer curves for the detection of TNP using **1** (**a**), **2** (**b**), and **3** (**c**): the inset shows the linear parts of the Stern–Volmer curves. Linear plots of the variation in emission intensity of **1** (**d**), **2** (**e**), and **3** (**f**) with respect to TNP concentration.

**Figure 6 sensors-25-00839-f006:**
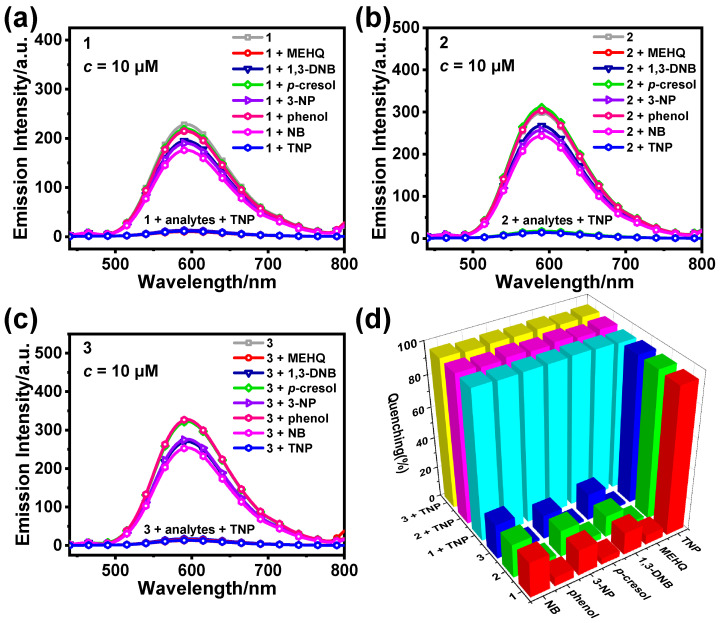
Emission spectra of **1** (**a**), **2** (**b**), and **3** (**c**) (10 μM) in CH_3_CN/H_2_O (*f*_w_ = 60%) after addition of different nitro compounds (20.0 equiv.). (**d**) Quenching rate of **1**–**3** (10 μM, CH_3_CN/H_2_O) for different nitro compounds (20.0 equiv.).

**Figure 7 sensors-25-00839-f007:**
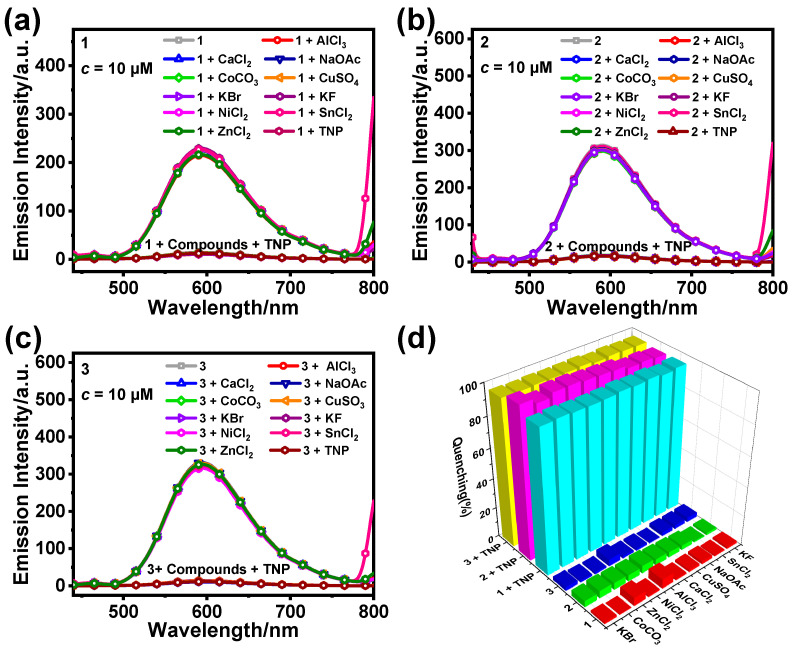
Emission spectra of **1** (**a**), **2** (**b**), and **3** (**c**) (10 μM) in CH_3_CN/H_2_O (*f*_w_ = 60%) after the addition of different ionic compounds (20.0 equiv.). (**d**) Quenching rates of **1**–**3** (10 μM, CH_3_CN/H_2_O) for different ionic compounds (20.0 equiv.).

**Figure 8 sensors-25-00839-f008:**
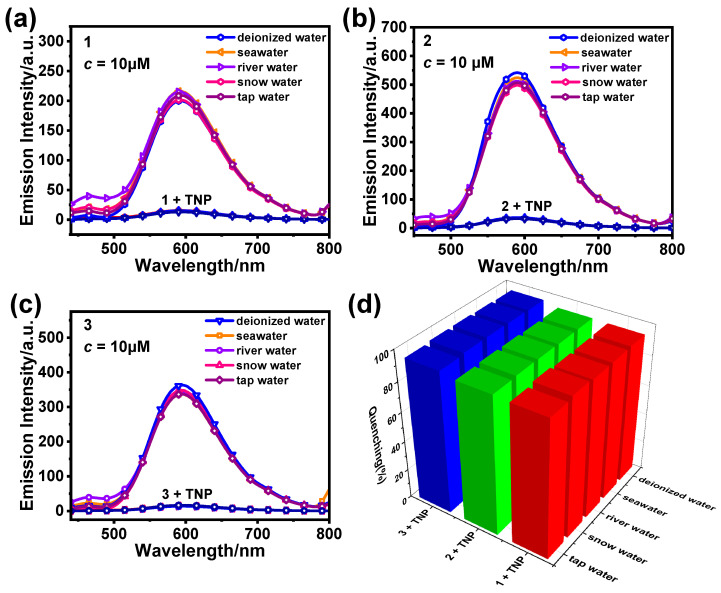
Emission spectra of **1** (**a**), **2** (**b**), and **3** (**c**) (10 μM) in CH_3_CN/H_2_O (*f*_w_ = 60%), using common water samples with or without TNP. (**d**) Quenching rates of **1**–**3** (10 μM, CH_3_CN/H_2_O) for TNP in different water sample detection systems.

**Figure 9 sensors-25-00839-f009:**
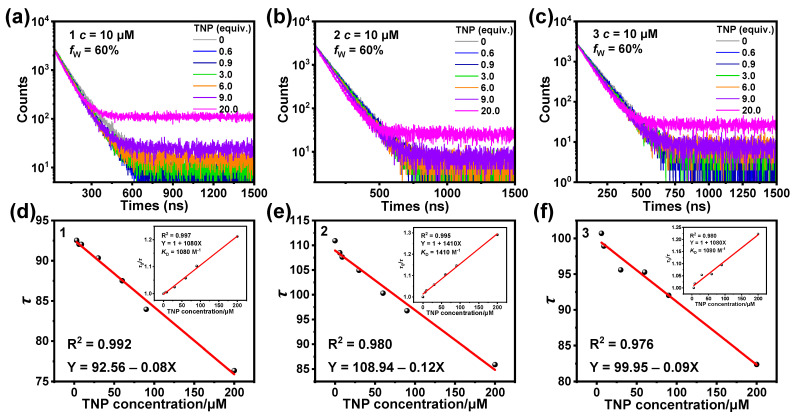
The lifetime decay traces of **1** (**a**), **2** (**b**), and **3** (**c**) after the addition of different concentrations of TNP in CH_3_CN/H_2_O. Changes in lifetime (*τ*) of **1** (**d**), **2** (**e**), and **3** (**f**) after the addition of different concentrations of TNP in CH_3_CN/H_2_O. Inset: Ratio of lifetime (*τ*_0_/*τ*) of **1**–**3** before (*τ*_0_) and after (*τ*) different concentrations of TNP were added.

**Figure 10 sensors-25-00839-f010:**
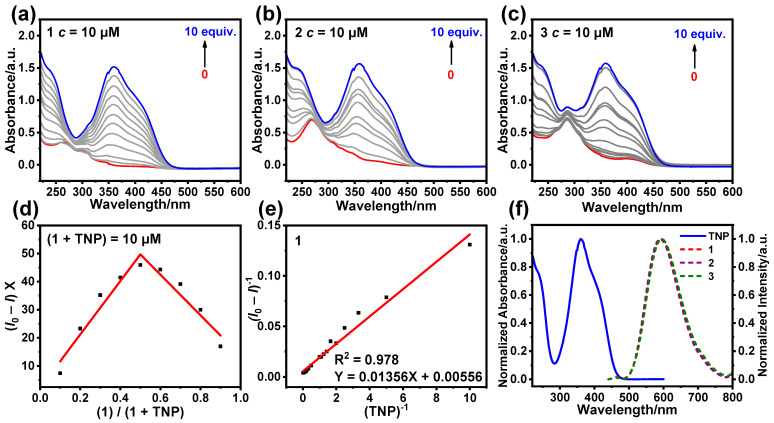
UV-Vis absorption spectra of **1** (**a**), **2** (**b**), and **3** (**c**) in CH_3_CN/H_2_O (*f*_w_ = 60%) after the addition of different concentrations of TNP. (**d**) The Job plot of **1** with TNP (**1**) represents the concentration of **1**; (**1** + TNP) represents the total concentrations of **1** and TNP; X represents the molar content of **1** in the mixed system of **1** and TNP; *I*_0_ and *I* represent the emission intensity of **1** before and after the addition of TNP, respectively.). (**e**) Benesi–Hildebrand plot of **1** with TNP ((TNP) represents the concentration of TNP; *I*_0_ and *I* represent the emission intensity of 1 before and after the addition of different concentrations of TNP, respectively.). (**f**) Normalized emission spectra of **1**–**3** (dashed line) and normalized UV-Vis absorption spectra of TNP (solid line).

## Data Availability

Data are contained within the article.

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
