# Peer review of "The Effect of the Position of a Phenyl Group on the Luminescent and TNP-Sensing Properties of Cationic Iridium(III) Complexes"

_sensors, 2025, doi:10.3390/s25030839_

Round 1

Reviewer 1 Report

Comments and Suggestions for Authors

In this manuscript, the authors have successfully synthesized a series of Ir() complexes exhibiting remarkable AIPE properties and sensitive detection of TNP. These materials demonstrate excellent resistance to common interfering substances and achieve a low detection limit in the nanomolar range. Despite the superior performance of this material system, the manuscript still requires significant revisions before it can be considered for publication in Sensors.

1.     In Figure 3, the emission intensity reaches a maximum when the water content is 60%, but is almost completely quenched with further increase in water content. This phenomenon is rather uncommon in AIE materials and requires a more detailed explanation from the authors. Additionally, in Figure 3b, there seems to be an underlying curve beneath the 70% data, and another purple curve appears without a corresponding label. The authors should carefully examine these anomalies.

2.     In addition to providing DLS analysis for sample 1 in Figure S4, the authors should also perform and present the same analysis for samples 2 and 3.

3.     For the organic compounds used in the interference experiments, such as NB, 1,3-DNB, MEHQ, and 3-NP, the full chemical names should be provided in addition to the abbreviations.

4.     For dynamic quenching, the Phosphorescence lifetime should decrease with increasing quencher concentration. However, Figure 9a shows the opposite trend. The Phosphorescence decay in this figure consists of at least two components (fast decay component and slow decay component). The slow decay component increases significantly with increasing quencher concentration, especially when TNP reaches 20 equiv. The authors need to distinguish these two components more carefully and conduct measurements over a longer time window.

5.     To maintain consistency in figure referencing, please add "Figure 10d" to lines 261-263 on page 11.

6.     The H NMR spectrum of the 1+TNP mixture in Figure S6 does not exhibit a pronounced shift in the TNP signals. To facilitate interpretation, the authors should explicitly label the exact chemical shifts of the relevant peaks. Furthermore, it is questionable whether such a minor perturbation in the NMR spectrum provides conclusive evidence for the proposed quenching mechanism. Given that structurally similar nitro compounds do not induce fluorescence quenching, it is intriguing to speculate whether these compounds would also fail to elicit noticeable changes in the NMR spectra?

7.     The proposed mechanism lacks sufficient detail. A more comprehensive explanation of the photo-induced electron transfer (PET) process is warranted. The inclusion of a Jablonski diagram would significantly enhance the clarity and understanding of the underlying photophysical processes.

Reviewer 2 Report

Comments and Suggestions for Authors

The article entitled "Effect of the position of a phenyl group on the luminescent and TNP-sensing properties of cationic iridium(III) complexes" fits well within the Aims and Scope of the journal Sensors. The authors present a study on chemical sensors based on three cationic Ir(III) complexes (1, 2, and 3) and their application in TNP (2,4,6-trinitrophenol) sensing. Furthermore, the authors describe the detection mechanism in detail.

In the abstract, the purpose of the study, the results obtained, and their potential applications are clearly summarized. The introduction is written in an engaging and accessible manner, encouraging readers to delve into the study. It effectively outlines the problem addressed by the research: the environmental contamination caused by TNP. The authors highlight the importance of detecting this harmful compound in aquatic environments to mitigate risks to human health. They discuss existing techniques for detecting nitro-explosive compounds and emphasize the advantages of photoluminescence and aggregation-induced emission (AIE) phenomenon. A review of iridium(III) complexes with AIE properties reported in the literature provides context for the current study, which is then outlined succinctly. The introduction successfully frames the research topic and establishes the need for this study.

The methodology is well-designed, with a logical and consistent presentation of the study’s steps. The research encompasses the synthesis and characterization of compounds 1–3, an investigation of their photophysical and aggregation-induced phosphorescence emission (AIPE) properties, TNP detection, and the sensing mechanism. The references include 41 sources, most published within the last five years, which are sufficient to contextualize the study and compare the results with existing literature.

I recommend the publication of the article with minor revisions.

1.     Absorption Characteristics: The absorption characteristics for compounds 2 and 3 in CH₃CN are missing from section 3.1 (Photophysical and AIPE properties). This information should be included in the article, with the spectra added to the Supplementary Data file. Reporting the absorption and emission maxima would provide a clearer picture of how the substitution position of the phenyl group influences the photophysical properties.

2.     Conclusions: The conclusions lack suggestions for future research directions. Including plans for further studies would strengthen the impact of the paper.

3.     Graphical Abstract: Adding a graphical abstract would enhance the article’s visibility and accessibility to a broader audience.

With these minor revisions, the article will make a valuable contribution to the field of chemical sensing and environmental monitoring.

Reviewer 3 Report

Comments and Suggestions for Authors

The article have reported three cationic Ir(III) complexes with a phenyl group at the pyridyl moiety in 2-phenylpyridine. All three complexes exhibit typical aggregation-induced phosphorescence emission (AIPE) properties in CH3CN/H2O. The AIPE property was further utilized to achieve highly sensitive detection of 2,4,6-trinitrophenol (TNP) in aqueous media with low limits of detection (LOD) of 164, 176, and 331 nM, respectively.  The article has potential to be accepted, but some issues should be clarified.

1、Update the literature. Please update some literatures in the past two years.

2、For complexes 2 and 3, the changes in the luminescent lifetime, proton nuclear magnetic resonance spectra and UV-Vis absorption spectra of before and after the addition of TNP should be measured.  

Round 2

Reviewer 1 Report

Comments and Suggestions for Authors

Thank you for your detailed response to my comments. Most of my questions have been satisfactorily addressed. However, considering the rigor of the conclusions, I would still suggest further elaboration on the PL decay mechanism in Figure 9. I understand the experimental challenges, and data beyond 20 equiv of TNP may not be strictly necessary. Nevertheless, I recommend extending the time scale, for instance, to 10 μs. Based on the current data, the PL decay can be divided into two phases. For example, in the case of 20 equiv, the fast decay component only lasts until approximately 400-500 ns. Moreover, with the increase of TNP, the proportion of the fast decay component gradually decreases while that of the slow decay component increases, suggesting that the complexes formed with the addition of TNP may exhibit slower PL decay. Additionally, I recommend maintaining consistent color schemes throughout the figures. The color scheme in Figure 9b differs from that in Figures 9a and 9c, which could potentially confuse the reader.
